# Genetic diversity and population structure analysis of soybean [*Glycine max* (L.) Merrill] genotypes based on agro-morphological traits and SNP markers

Abebawork Tilahun Assfaw[1,2]*, Olasanmi Bunmi[1,3], Agre Paterne[4], Godfree Chigeza[4], Hapson Mushoriwa[4], Kayode Fowobaje[4], Abush Tesfaye Abebe[4]

**1** Pan African University Life and Earth Science Institute (including Health and Agriculture), Ibadan, Nigeria, **2** Hawassa University College of Agriculture, Hawassa, Ethiopia, **3** Department of Crop and Horticultural Sciences, University of Ibadan, Ibadan, Nigeria, **4** International Institute of Tropical Agriculture, Ibadan, Nigeria

* abebaworktilahun@gmail.com

## Abstract

Soybean (*Glycine max*) is one of the world's most important oilseed crops and has adapted to various environmental conditions. Yields of soybeans in Nigeria are notably low due to different production constraints, including the limited availability of improved varieties and the slow replacement rate of old varieties with new and high-yielding ones. Ensuring high genetic diversity in the working germplasm is among the primary factors for the success of breeding programs in identifying high-yielding and well-adapted improved varieties. This study aimed to assess the genetic diversity and population structure of 45 soybean breeding lines of the International Institute of Tropical Agriculture soybean breeding program at the advanced evaluation stage based on phenotypic traits and SNP markers to support breeding strategies. Field trials were conducted in 2022 across three International Institute of Tropical Agriculture stations in Nigeria using a 5 × 9 alpha-lattice design with three replications. The collected yield and yield component data were subjected to analysis of variance, mean comparison, principal component analyses, and cluster analyses using R software. The genotypes were further assessed using 10,630 SNP markers obtained from DArTseq genotyping. The combined analysis of variance revealed a significant genotype × location interaction for grain yield and a highly significant difference in days to 50% flowering and days to 95% maturity. The genotypes G02, G10, G11, G01, and G24 were significantly superior in grain yield. Principal component analysis showed that the first three components explained 64.8% of total variation, with major contributions from traits such as lodging score, hundred seed weight, plant height, nodulation, and days to 50% flowering. Hierarchical clustering grouped the genotypes into five clusters, highlighting desirable traits such as high yield, early maturity, and lodging tolerance. SNP-based population structure grouped

**Data availability statement:** The data underlying the results presented in the study are available from IITA Data Bank, https://data.iita.org/ with the associated DOI, https://doi.org/10.25502/mja8-g609/d.

**Funding:** This research was funded by the Pan African University Life and Earth Science Institute (including Health and Agriculture) (PAULESI) as part of the MSc funding awarded to ATA with grant number PAU020110MB (https://www.pau-au.africa/instiues/paule-si). The study was also supported by the International Institute of Tropical Agriculture (IITA) in connection with the United States Agency for International Development (IITA/USAID) Genetic Improvement in Soy Project with number PJ-2315 (https://www.iita.org/iita-countries/nigeria/). The funders had no role in study design, data collection and analysis, decision to publish, or preparation of the manuscript.

**Competing interests:** The authors have declared that no competing interests exist.

the genotypes into three distinct subpopulations. The SNP markers showed average observed heterozygosity, expected heterozygosity, minor allele frequency, and polymorphic information content of 0.08, 0.27, 0.20, and 0.22, respectively, which showed the existence of considerable genetic variation among the studied genotypes.

## Introduction

Soybean [*Glycine max* (L.) Merrill] is one of the leading oilseed crops globally, accounting for approximately 57% of global vegetable oil production [1,2]. It is cultivated in numerous countries and serves as a major source of vegetable oil and protein, essential in food, feed, and various industrial applications [3]. With a protein level of 40–42% and an oil content of 18–22%, soybeans have twice as much protein as meat or chicken and all eight essential amino acids required for a child's healthy development [4]. Soybeans were domesticated around the 11th century BC in Northeast China and subsequently spread across Asia, the USA, Brazil, and Argentina [5]. Nigeria ranks as the second-largest producer of soybeans in Africa, following South Africa [6]. Soybeans were first introduced to Nigeria in 1908, with successful commercial cultivation beginning in 1937, using the Malayan variety in Benue State [7]. The crop is adapted to diverse environmental conditions and is predominantly grown under rain-fed conditions [8]. Despite Nigeria being the second highest producer of soybeans in Africa, the national average yield is < 1 ton/ha, which is far below the potential yield of the crop (over 3 tons/ha) [6]. However, the limited availability of high-yielding and disease-resistant varieties and the slow rate of replacement of old varieties with high-yielding, climate, and stress-resilient varieties are among the major factors contributing to the low yields of soybeans. Hence, enhancing the genetic improvement of the crop to make high-yielding, climate-resilient varieties available for production can have paramount importance in increasing soybean production, which will greatly benefit humanity, primarily by reducing malnutrition. Genetic improvement of soybeans plays a pivotal role in addressing malnutrition in Nigeria through enhancing the crop's nutritional composition [9]. Currently, efforts are underway by the IITA soybean breeding program to improve protein content, fatty acid profiles, and reduce anti-nutritional factors such as phytic acid and protease inhibitors, thereby increasing the crop's nutritional value and its role in combating malnutrition. Recent studies have shown that local soybean varieties differ significantly in their proximate composition, including protein and fat levels, highlighting the potential for selecting superior nutritional lines [10,11]. Additionally, integrating biofortification strategies into soybean breeding can help enhance the supply of essential micronutrients like iron and zinc, which are commonly deficient in low-income populations [12,13]. Thus, soybean improvement strengthens food security and serves as a tool for mitigating micronutrient deficiencies and protein-energy malnutrition [14,15].

A fundamental step for success in any breeding program is evaluating and understanding the extent of genetic variability in the crop of interest [16]. The genetic diversity of a crop species can be assessed using phenotypic traits and molecular

markers [17]. A phenotypic diversity study is the standard method of evaluating the extent of genetic diversity and determining the agronomic value and grouping of crop germplasm [18]. Understanding the phenotypic variation and trait relationships helps crop breeders to develop more adaptable and productive varieties [17]. Key traits, like number of seeds per plant/pod, number of pods per plant, 100 seeds weight, leaflet shape, flower color, stem architecture, number of days till flowering or maturity, plant height, pubescence type, and density, grain yield and other related factors are among the phenotypic qualities that are typically evaluated for genetic diversity of soybean [1 9–25]. Several studies on phenotypic traits have found a high genetic diversity in soybean germplasms. Liu et al. [26] reported high phenotypic variation in characterizing 138 soybean accessions based on yield and yield-related agro-morphological traits. Similarly, Bairagi et al. [27] reported high genetic variation among 32 soybean genotypes based on ten morphological traits, while Kuswantoro et al. [28] studied the phenotypic diversity of 100 soybean genotypes and reported significant variations for all the agronomic traits. Based on nine agro-morphological traits, Marconato et al. [29] also reported the existence of high genetic diversity among the 93 soybean accessions maintained by the Brazilian Agricultural Research Corporation (EMBRAPA) gene bank. All the aforementioned findings indicate that the agro-morphological traits were helpful in assessing genetic diversity that facilitates the utilization of the genetic resources for the genetic improvement of the crop. In crop species and their relatives, selection based on phenotypic features is still frequently used and will likely continue to be an important approach in determining the extent of diversity [25].

Molecular markers are the preferred approach for assessing genetic diversity due to their excellent repeatability, superior genome coverage, automation potential, great variability, neutrality, and lack of sensitivity to environmental variations [30]. There are reports on using different types of molecular markers for diversity and population structure studies in soybeans. However, SNP markers are the most commonly used molecular markers in genetic diversity in most of the recent studies, given that they are extensively spread across the plant's genome [31]. This is due to its affordability, target accuracy, and codominant character [32,33]. Genetic diversity studies based on SNP markers were conducted on various crops including soybeans [34–37] cassava [33]; maize [38–40]; and yam [41–43]. Despite, the great significance of assessing the phenotypic and genotypic diversity, the study materials that were developed by the IITA soybean breeding program and were at an advanced stage of evaluation have not been assessed for their genetic diversity based on both the phenotypic traits and molecular markers. Therefore, the objectives of this study were to evaluate the genetic diversity and population structure of the advanced soybean breeding lines of the IITA soybean breeding program for yield and yield-related traits using agro-morphological traits and SNP markers to recommend the best-performing varieties for direct production or use as parental lines for future genetic improvement of the crop.

## Materials and methods

### Description of the study area

The field experiments were conducted across three stations, i.e., Ibadan, Zaria, and Ikenne of the International Institute of Tropical Agriculture (IITA), Nigeria in the 2022 cropping season. The study locations represent Nigeria's different soybean production environments and are characterized by different agroclimatic conditions presented in Table 1.

**Table 1. Trial locations and their respective agro-climatic descriptions.**

| Location | Ecology | State | Long. | Lat. | Elevation (masl) | Rainfall (mm) | Temp.(°C) | |
|---|---|---|---|---|---|---|---|---|
| | | | | | | | Min | Max |
| IITA, Ibadan | Derived Savannah | Oyo | 7°30′N | 3°54′E | 243 | 1300-1500 | 22 | 32 |
| Ikenne | Lowland humid forest | Ogun | 6°52′N | 3°43′E | 235.2 | 1200 | 24 | 30 |
| Zaria | Northern Guinea. Savannah | Kaduna | 11°11′N | 7°38′ E | 600 | 1045 | 17.1 | 35.4 |

masl = meter above sea level, mm = millimeter,°C = degree Celsius Source: [44,45].

## Experimental materials

A total of 45 soybean breeding lines, which are part of the working germplasm of the IITA soybean breeding program, along with the IITA check (TGX-1951-3F a variety developed by IITA and released in Nigeria) and a commercial check (SC-Signa, a variety released by a private company called SeedCo in Nigeria) were used in the study. Among the entries, 38 genotypes were developed by and sourced from the soybean breeding program of IITA that were at an advanced stage of yield trials in Nigeria; four from the USDA soybean genetic resource center, one from Ghana, and one from Uganda. The study genotypes with their corresponding sources are presented in Table 2.

## Experimental design and management

The field trial was laid out in a 5 × 9 alpha lattice design with three replications. Each entry was planted in a plot of 4 rows of 4 m length. The spacings between rows and plants were 50 cm and 5 cm, respectively. The two middle rows were harvested to measure plot yield and other related traits, and two border rows were left to exclude the border effect. A mixture of NPK and TSP fertilizers was applied at 25g/row at a 1:2 ratio at planting. The seeds were inoculated with *Bradrhizobium japonicum* inoculant called Nodumax, manufactured by the IITA Business Incubation Platform (BIP). All the rest of the management practices were applied as per the recommendation for the crop [46].

## Phenotypic data collection

The agro-morphological traits including plant height and root nodule score were determined from the average values of five randomly selected plants of each genotype, whereas days to 50% flowering, days to 95% maturity, lodging score, shattering score, hundred seed weight and grain yield were collected on a plot basis from the trials following the soybean descriptor of IBPGR [47] as shown in Table 3.

## Phenotypic data analysis

**Analysis of variance and mean comparisons.** The quantitative data were subjected to a combined analysis of variance to test for significant differences among genotypes using the linear mixed model (LMM) procedure of the R software package (version 4.3.1, 2023). Locations and replications within locations were considered random effects, whereas the genotypes were considered as fixed effects and used to determine the significance level of genotypes (G), environments (E), and their interaction (GEI). The combined ANOVA model used in this study is provided in the following equation.

$$Y_{ijkl} = \mu + G_i + E_j + Rk_{(j)} + BI_{(jk)} + GE_{ij} + e_{ijlk}$$

where $Y_{ijkl}$ is the response of the $i^{th}$ genotype in $j^{th}$ environment and $k^{th}$ replication within the $j^{th}$ environment and $l^{th}$ block within replication; μ is the grand mean, $G_i$ is the effect of $i^{th}$ genotype; $E_j$ is the effect of $j^{th}$ environment; $R_{k(j)}$ is the effect of $k^{th}$ replication within the $j^{th}$ environment; $B_{l(jk)}$ is the effect of $l^{th}$ block in the $j^{th}$ environment and $k^{th}$ replication; $GE_{ij}$ is the interaction effect of ith genotype and $j^{th}$ environment; and $e_{ijkl}$ is the random error effect.

The mean comparisons were done using the least significant differences (LSD) at a 5% level of significance.

**Cluster analysis.** Cluster analyses were used to group the genotypes into homogeneous forms based on quantitative characters. A dissimilarity matrix was first computed using Euclidean distance, which is appropriate for continuous quantitative traits. Hierarchical clustering was then performed using Ward's D² method (implemented in R as ward.D2), based on the dissimilarity coefficients among the 45 soybean genotypes. The analysis was performed using the base R function, and the dendextend package was used only to visualize the dendrogram (version 4.3.1, R Core Team, 2023).

**Principal components analysis (PCA).** Principal component analysis (PCA) was computed to determine the traits that accounted for much of the total variation and to assess the extent of genetic diversity in the studied genotypes. The analysis was performed using the 'FactoMiner' package for PCA and the 'factoextra' package for visualization in

**Table 2. The pedigrees, code, and source of 45 soybean genotypes were used in the study.**

| No | Pedigree | Genotype code | SNP designation | Source |
|---|---|---|---|---|
| 1 | TGx1951-4F | G01 | SY001 | IITA |
| 2 | TGx1993-18FN | G02 | SY002 | IITA |
| 3 | TGx2015-2E | G03 | SY003 | IITA |
| 4 | TGx1989-11FxTGx1987-10F-1-1-3-1-3-I | G04 | SY004 | IITA |
| 5 | TGx1961-1FxH10-2-3-7-2-1 | G05 | SY005 | IITA |
| 6 | TGx1961-1FxTGx1835-10E-1-1-4-2-1-E | G06 | SY006 | IITA |
| 7 | TGx1987-9FxTGx1835-10E-1-2-2-1-1 | G07 | SY007 | IITA |
| 8 | TGx1987-11FxH7-3-1-1-1-2-3-I | G08 | SY008 | IITA |
| 9 | TGx1987-11FxH7-3-1-1-1-2-6-I | G09 | SY009 | IITA |
| 10 | TGx1989-19FxTGx1987-10F-5-3-1-2-2-I | G10 | SY010 | IITA |
| 11 | TGx1989-45FxTGx1835-10E-3-2-1-3-3-E | G11 | SY011 | IITA |
| 12 | TGx2022-3E | G12 | SY012 | IITA |
| 13 | TGx1987-62FxH7-1-1-4-3-2-E | G13 | SY013 | IITA |
| 14 | TGx1990-38FxTGx1835-10E-1-4-3-1-1-E | G14 | SY014 | IITA |
| 15 | TGx2029-21F | G15 | SY015 | IITA |
| 16 | Panaroma-3 | G16 | SY016 | USDA |
| 17 | TGx2029-7F | G17 | SY017 | IITA |
| 18 | Panaroma-1 | G18 | SY018 | USDA |
| 19 | TGx2029-31F | G19 | SY019 | IITA |
| 20 | TGx2029-49F | G20 | SY020 | IITA |
| 21 | Panorama-27D | G21 | SY021 | USDA |
| 22 | TGx2029-53F | G22 | SY022 | IITA |
| 23 | TGx2014-16FM | G23 | SY023 | IITA |
| 24 | Sc-Signa | G24 | SY024 | Seedco |
| 25 | Songda | G25 | SY025 | SARI, Ghana |
| 26 | TGx2029-39F | G26 | SY026 | USDA |
| 27 | TGx1951-3F | G27 | SY027 | IITA-Check |
| 28 | (TGx1987-9F/TGx1740-2F)-#F5-1025–1 | G28 | SY028 | IITA |
| 29 | (TGx1740-2F/MW1)-#F6-2002-10-15 | G29 | SY029 | IITA |
| 30 | (TGx1740-2F/MW1)-#F6-2002-10-19 | G30 | SY030 | IITA |
| 31 | (TGx1740-2F/MW1)-#F6-2002-10-28 | G31 | SY031 | IITA |
| 32 | (TGx1830-20E/UG5)-#F6-2014-27-5 | G32 | SY032 | IITA |
| 33 | (TGx1830-20E/UG5)-#F6-2014-27-6 | G33 | SY033 | IITA |
| 34 | TGx2029-22F | G34 | SY034 | IITA |
| 35 | TGx1448-2E | G35 | SY035 | IITA-check |
| 36 | TGx1989-19F | G36 | SY036 | IITA-check |
| 37 | (TGx1740-2F/MW1)-#F6-2002-10-11 | G37 | SY037 | IITA |
| 38 | TGx2029-20F | G38 | SY038 | IITA |
| 39 | TGx2029-27F | G39 | SY039 | IITA |
| 40 | (TGx1987-10F/TGx1740-2F)-#F5-1011–6 | G40 | SY040 | IITA |
| 41 | (TGx1987-10F/TGx1740-2F)-#F5-1011–8 | G41 | SY041 | IITA |
| 42 | (TGx1987-62F/MW1)-#F5-1006–15 | G42 | SY042 | IITA |
| 43 | (TGx1987-62F/MW1)-#F5-1006–22 | G43 | SY043 | IITA |
| 44 | Maksoy-4N | G44 | SY044 | Uganda |
| 45 | (TGx1740-2F/MW1)-#F6-2002-10-21 | G45 | SY045 | IITA |

**Table 3. List of traits and their measurements.**

| SN | Traits | Code | Measurement | Nature of the traits |
|----|--------|------|-------------|----------------------|
| 1 | Plant height (cm) | PH | measured in cm using a centimeter ruler as the height of the plant from the base to the tip of the plant at maturity | Quantitative |
| 2 | Days to 50% flowering (days) | D50F | The number of days taken from sowing to 50% of the plant bears at least one flower | Quantitative |
| 3 | Days to 95% maturity (days) | D95M | Recorded when at least 95% of the pods attained a mature brown color. | Quantitative |
| 4 | Lodging (score) | Lodg | All plants in the middle two rows were scored using a scale of 1–5; where 1 = all plants erect, 2 = slight lodging, 3 = plants lodged at a 45-degree angle, 4 = severe lodging, and 5 = all plants flat. | Quantitative |
| 5 | Root Nodule (score) | RtNd | Plants from the border rows were uprooted and visually scored using a scale of 1–5; where 1 = no nodules, 2=with few nodules, 3 = half of the roots have nodules, 4=more than half of the roots have nodules, and 5 = all roots have nodules | Quantitative |
|  | Shattering (score) | SHS | was scored from the border rows, one week after harvesting the middle rows, using a scale of 1–5; where 1 = very tolerant, 2 = tolerant, 3 = moderately tolerant, 4 = highly susceptible, and 5 = very highly susceptible | Quantitative |
| 7 | Hundred seed weight (g) | HSW | The weight of a hundred seeds selected at random from each of the genotypes was recorded and expressed in grams | Quantitative |
| 8 | Grain yield (kg) | GY | Seeds from plants harvested from the middle two rows and contributing to the yield sample in the net plot were subjected to uniform drying conditions, weighed in grams, and converted to kilograms per hectare. | Quantitative |

R software [48]. In this analysis, only principal components (PCs) with eigenvalues greater than one were considered important for the total variations. PCA was not used to select traits for clustering.

## Genotyping

**Leaf sampling, DNA extraction, and genotyping data processing.** Seeds of the 45 genotypes were sown in the screen house at IITA, Ibadan, for sampling. For the analysis, fully expanded but young leaves from three-week-old seedlings were collected from four to five plants of each of the 45 genotypes (Table 2) using a leaf puncher and kept in a zip-lock bag on ice and later stored at −80 ◦C in a deep freezer dryer. Before genomic deoxyribonucleic acid (DNA) extraction, each sample leaf was bulked and lyophilized for 72 hours in a Labconco Freezone 2.5L System lyophilizer (Marshall Scientific, LABCONCO, Kansas, MO, USA) and reduced to a fine powder in the SpexTM Sample Prep 2010 Geno/Grinder (Thomas Scientific, Metuchen, NJ, USA). The deoxyribonucleic acid (DNA) was extracted using a technique developed by Intertek-AgriTech (http://www.intertech.com/agriculture/agritech/), accessed 16 January 2024, and based on the LGC oKtopure™ automated high-throughput 'sbeadex™' DNA extraction and purification system (https://www.biosearchtech.com/), accessed 16 January 2024.

Magnetic separation was used in the 'sbeadex™' technique to prepare nucleic acids. The first stage in this process was to homogenize leaf tissue samples in 96 deep-well plates using steel bead grinding. The ground tissue was treated with a DNA extraction buffer using LGC's 'sbeadex™' kit for plant DNA preparation (https://www.biosearchtech.com/, accessed 16 January 2024). Finally, super-paramagnetic particles coated with 'sbeadexTM' surface chemistry that catches nucleic acids from a sample were used to purify extracted DNA. Purified DNA was eluted and used in downstream procedures. Medium-throughput genotyping was conducted in a 96 plex DArTseq protocol, and SNPs were called using the DArT's proprietary software, DArTSoft, as described by Kilian et al. [49]. Each sequencing result's reads and tags were mapped to the G. max reference genome, which was used to convert the raw HapMap file to a Variant Call Format (VCF).

**Genotypic data analysis.** A total of 59,126 SNP markers were identified from the raw DArTseq SNP-derived dataset before quality assessment. VCFtools [50] was used to perform the initial filtering, which involved removing SNP markers with a minor allele frequency (MAF) < 0.01, markers with >20% missing data (i.e., SNP call rate < 80%), unmapped markers to any chromosome, and duplicated markers. Subsequently, PLINK v1.9 was employed for additional quality control, specifically for excluding SNPs with high heterozygosity using Hardy–Weinberg Equilibrium (HWE)-based filtering. In the end, 10,630 informative SNP markers were retained and used for the subsequent analysis. Diversity indices statistics, such as observed heterozygosity (Ho), expected heterozygosity (He), minor allele frequency (MAF), and the polymorphic information content (PIC) were estimated using PLINK 1.9 [51]. Ho was calculated with the method suggested by Chesnokov and Artemyeva [52]:

$$H_0 = \frac{\text{Total number of heterozygote individuals}}{\text{Sample size}}$$

Equation 1

He was determined following the equations given by Liu [53]

$$H_e = 1 - \sum pi^2$$

Equation 2

Where the summing is over all possible alleles, and **pi** is the frequency of the $i^{th}$ allele
The MAF values were calculated using Xue et al. [54] equation as follows:

$$MAF = \frac{X_i}{X}$$

Equation 3

where $X_i$ is the number of minor alleles detected at a point, and X is the total number of genotypes detected at a point. In the same way, the PIC values were calculated by the following formula: Amiryousefi et al. [55]

$$PIC = 1 - \sum pi^2 - \sum \sum pi^2 pj^2$$

Equation 4

Where pi and pj represent the population frequencies of the i-th and j-th alleles, respectively. The first summation includes all alleles, while the double summation covers all combinations, where $i \neq j$.

Bayesian information criterion (BIC) was used to define the optimum sub-populations (K) using discriminant analysis of principal components (DAPC) and which was implemented in R using the 'adegenet' package [56]. Using the ancestry probability, the level of admixture was estimated, and individuals were assigned to a specific population when their membership coefficient in that group was ≥ 0.70. Genotypes with membership coefficients less than 0.70 at each assigned K were considered admixed. Coefficients of similarity showing genetic distances among the soybean lines were calculated in R software following Gower's Distance model [57].

## Results

### Combined analysis of variance for agronomic traits over locations

The results from the combined analysis of variance (ANOVA) across three locations are presented in Table 4. The combined ANOVA revealed a significant (p ≤ 0.05) genotype × location interaction for grain yield and a highly significant (P < 0.001) difference for days to 95% maturity and days to 50% flowering. However, a non-significant genotype x location interaction was observed for shattering, root nodules, lodging, plant height, and hundred seed weight. The genotype effect was highly significant (p ≤ 0.001) for all the traits except for hundred seed weight. The location effect also showed a significant effect (P < 0.05) for most traits, except for lodging, days to 50% flowering, and hundred seed weight.

**Table 4. Mean squares for eight agronomic and yield traits of 45 soybean genotypes evaluated in three agroecologies of Nigeria.**

| SOV | Genotype (G) | Location(L) | G×L | Error 1 | Error 2 |
|---|---|---|---|---|---|
| DF | 44 | 2 | 88 | 6 | 72 |
| SHS | 1.13*** | 4.25* | 0.39ns | 0.00 | 0.00 |
| RtNd | 1.14*** | 71.3*** | 0.68ns | 0.01 | 0.03 |
| Lodg | 1.2*** | 0.22ns | 0.41ns | 0.06 | 0.08 |
| PH | 839*** | 1010*** | 138ns | 0.00 | 9.34 |
| D50F | 42.7*** | 27.7ns | 34.9*** | 0.00 | 0.20 |
| D95M | 37.3*** | 376*** | 88*** | 0.98 | 0.80 |
| HSW | 93ns | 126ns | 76.2ns | 0.31 | 0.00 |
| GY | 846000*** | 7900000*** | 501000* | 18177.13 | 86991.55 |

*, *** Significant difference at p < 0.05 and p < 0.001, ns = non-significant, SOV = source of variation, DF = degree of freedom, GxL = genotype by location interaction, SHS = shattering score, RtNd = root nodule score, Lodg = lodging score, PH = plant height in centimetre, D50F = days to 50% flowering, D95M = days to 95% maturity, HSW = hundred seed weight in gram, GY = grain yield in kilogram per hectare.

### Location-specific performance of superior genotypes

Given the significant genotype×environment (G×E) interaction, the best-performing genotypes for each location were identified based on grain yield, days to 50% flowering, and days to 95% maturity. At Ibadan, genotype **G24** recorded the highest grain yield (3,910 kg/ha), while **G10** and **G11** exhibited the earliest flowering (47 days) and maturity (109 days), respectively. At Zaria, genotype **G05** achieved the highest yield (3,070 kg/ha), whereas genotypes **G23** and **G32** were the earliest to flower (37.9 days) and mature (102 days), respectively. At Ikenne, genotype **G36** had the highest yield (3,850 kg/ha), while **G32** and **G03** recorded the earliest flowering (47.2 days) and maturity (124 days), respectively. Detailed values are provided in **S5** Table.

### Mean performances of the soybean genotypes for yield and yield-related traits

The mean comparisons of agronomic and yield-related traits that were determined during the study are presented in **S3** Table. High mean yields of 3310, 3210, 3060, 3050, and 2990 kg/ha were recorded for genotypes G02, G10, G11, G01, and G24, respectively. The lowest-yielding genotypes were G35 (1700 kg/ha), G15 (2020 kg/ha), G18 (2040 kg/ha), and G28 (2140 kg/ha). Days to 95% maturity varied from 112 to 121 days with G32 and G33 maturing earlier (112 days) than the rest, while G25 was relatively late maturing (121 days). Mean days to maturity value of 117 days were recorded for genotypes G02, G26, G36, G39, G40 and G45. Days to 50% flowering varied from 44 to 55 days with the earliest genotype being G23 (44 days) followed by G33 with 45 days and G24 and G39 with 46 days. The genotypes that took longer to flower were G04, G05, G08, and G13 all with 55 days. Shattering ranged from 0.96 to 3.5 with an average of 1.3. Genotype G45 had a higher shattering score of 3.5. Lower shattering scores were recorded by G43 (0.96), G27, G40, and G07 (0.97). The maximum lodging score was recorded by G44 (2.43), G01 (2.4), and G08 (2.3), while the minimum was observed in G22 (1), G20 (1.02), and G17 (1.06). The root nodule score ranged from 2.17 to 3.61 with an average of 3.13. Among the 45 soybean genotypes, 27 genotypes nodulated more efficiently (above average) and the best were genotypes G02, G16 (3.61), G24, G29, G45 (3.56), and G01, G07, G15 (3.50). Genotypes G17 (102 cm), G20 (98.6 cm), G16 (96.4 cm), G19 (95.2 cm) and G26 (94.3 cm) were found to be the tallest. Genotypes G02 (63.6 cm), G03 (64.8 cm), G31 (67.4 cm), and G12 (68.3 cm) were the shortest genotypes in this study.

### Principal component analysis (PCA)

The first three principal components with eigenvalues greater than one contributed to 64.8% of the total variation among the genotypes (Table 5, **S1** Table). The first principal component (PC1) accounted for 27.2% of the total variation. The lodging score showed a high and positive association, while days to 95% maturity had a negative association with PC1.

**Table 5. Eigenvalues, percent of the variance, and cumulative variance of the first three PCs of the studied soybean genotypes for eight agronomic and yield traits.**

| Traits | PCs | | |
|---|---|---|---|
| | I | II | III |
| | Eigenvectors | | |
| D50F | −0.13 | −0.53 | 0.51 |
| RtNd | −0.25 | 0.27 | 0.65 |
| Lodg | 0.42 | −0.38 | 0.21 |
| SHS | 0.26 | 0.23 | 0.38 |
| PH | −0.39 | 0.42 | 0.04 |
| D95M | −0.53 | −0.11 | 0.17 |
| HSW | 0.33 | 0.51 | 0.12 |
| GY | 0.38 | 0.10 | 0.29 |
| Eigenvalues | 2.17 | 1.71 | 1.30 |
| Partial variation (%) | 27.2 | 21.3 | 16.3 |
| Cumul. variation (%) | 27.2 | 48.5 | 64.8 |

PCs = Principal components, SHS = shattering score, RtNd = root nodule score, Lodg = lodging score, PH = plant height (cm), D50F = days to 50% flowering, D95M = days to 95% maturity, HSW = hundred seed weight (gm), GY = grain yield (kg/ha).

The considerable variation observed in the second principal component (21.3%) was positively and highly correlated with hundred seed weight and plant height, and negatively correlated with days to flowering. The third principal component accounted for 16.3% of the total variation and had a high positive correlation with nodulation and days to 50% flowering. The biplot based on PC1 and PC2 that captured 27.2 and 21.3% of the total variation, respectively, with a cumulative contribution of 48.7% (Fig 1) displayed that genotypes G15, G24, G45, G13, and G02 were far from the origin, whereas G23, G43, G11, G07, G01, and G09 were located close to the origin.

## Cluster analysis

The cluster analysis resulting from the hierarchical ascending classification (HAC) grouped the soybean genotypes into five cluster groups, as shown in Table 6 and Fig 2.

Cluster I had the highest number of genotypes (12); while cluster IV had the lowest number of four genotypes. F-test revealed significant differences among the five clusters for grain yield, plant height, and lodging score ($p < 0.001$), and for days to 95% maturity ($p < 0.01$). Cluster IV exhibited the highest mean grain yield (2786.85 kg/ha), while Cluster III recorded the lowest (2319.34 kg/ha), indicating meaningful yield differences across clusters. For days to 95% maturity, Cluster V was the latest maturing group (116 days), significantly differing from Cluster II, the earliest group (109 days), suggesting potential suitability for diverse growing seasons. Lodging score also varied significantly, with Cluster V showing the lowest lodging score (1.23), making it desirable for environments prone to lodging. Similarly, plant height differed significantly across clusters, with Cluster V being the tallest (92.10 cm) and Cluster II the shortest (70.92 cm). In contrast, differences among clusters for days to 50% flowering, hundred seed weight, root nodule score, and shattering score were not statistically significant based on F-test.

## Genetic diversity of soybean displayed by SNP markers

The results of the SNP markers' statistics are summarized in Table 7 and S1 Fig. The observed heterozygosity ranged from 0.00 to 0.16, with a mean of 0.08, while the expected heterozygosity (He) ranged between 0.00 and 0.50, with a mean of 0.27. The minor allele frequency (MAF) ranged between 0.01 and 0.50 with a mean value of 0.20 and the polymorphic information content (PIC) ranged from 0.00 to 0.38 with an average value of 0.22.

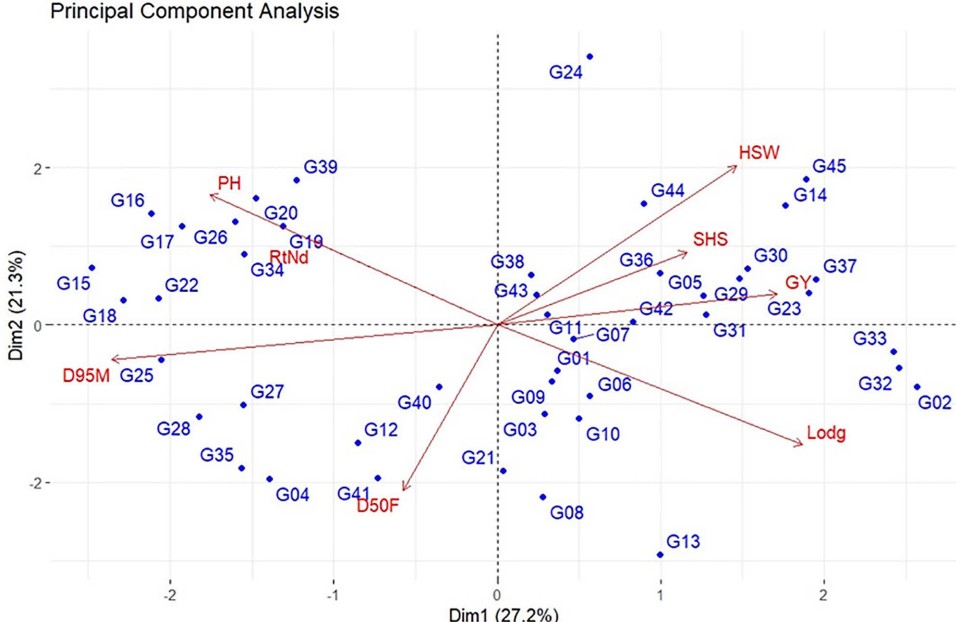

**Fig 1. Distribution of genotypes and traits in PCA-Biplot of the soybean genotypes evaluated for eight agronomic and yield-related traits.** Where SHS = shattering score, RtNd = root nodule score, Lodg = lodging score, PH = plant height (cm), D50F = days to 50% flowering, D95M = days to 95% maturity, HSW = hundred seed weight (gm), GY = grain yield (kg/ha).

**Table 6. Cluster mean values of the studied agro-morphological traits of the soybean genotypes and their clustering, including least significant difference (LSD) values and significance levels based on F-test.**

| Traits | Clusters | | | | | LSD | Sign |
|---|---|---|---|---|---|---|---|
| | I (12) | II (8) | III (10) | IV (4) | V (11) | | |
| D50F | 52 | 47 | 53 | 48 | 47 | 4.47 | ns |
| D95M | 112 | 109 | 115 | 112 | 116 | 1.28 | ** |
| GY (kg/ha) | 2695.49 | 2688.74 | 2319.34 | 2786.85 | 2357.05 | 173.10 | *** |
| HSW (g) | 15.07 | 15.31 | 12.99 | 16.18 | 14.00 | 2.05 | ns |
| Lodg | 2.05 | 1.89 | 1.76 | 1.49 | 1.23 | 0.21 | *** |
| PH (cm) | 81.35 | 70.92 | 73.09 | 76.73 | 92.10 | 5.38 | *** |
| RtNd | 3.33 | 2.67 | 3.06 | 3.33 | 3.24 | 0.23 | ns |
| SHS | 1.25 | 1.22 | 1.16 | 2.36 | 1.17 | 0.56 | ns |

NB: Numbers in parenthesis are the number of genotypes in that cluster, D50F = days to 50% flowering, D95M = days to 95% maturity, GY = grain yield in kilogram per hectare, HSW = hundred seed weight in gram, Lodg = lodging score, PH = plant height in centimeter, SHS = shattering score, RtNd = root nodule score, LSD = least significant difference, **, *** Significant difference at p < 0.01 and p < 0.001, ns = non-significant.

The population structure analysis in this study identified three sub-populations among the 45 soybean genotypes based on the optimal K = 3 determined based on the BIC method (Fig 3).

Using the 70% cut-off criterion of the membership probability threshold, 27 genotypes were successfully assigned to the three different sub-populations. The remaining 18 genotypes with a probability of associations of less than 70% were considered as an admixed population (Fig 4, **S2** Table).

Sub-population I (red) was comprised of 13 genotypes that were sourced from IITA and Uganda, while the seven genotypes were allocated in sub-population II (blue) that were sourced from Ghana and IITA. Sub-population III (green)

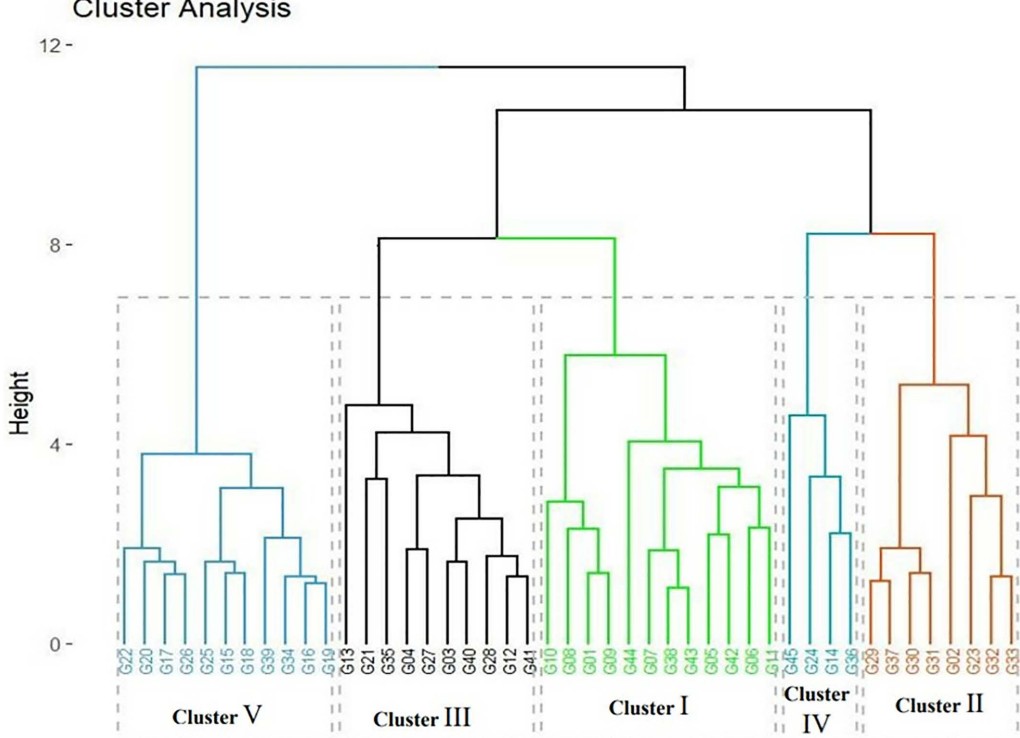

**Fig 2. Dendrogram of the studied soybean genotypes evaluated across three locations in Nigeria.**

**Table 7. Diversity indices statistics of the 45 soybean genotypes based on 10,630 SNP markers.**

|  | Ho | He | MAF | PIC |
|---|---|---|---|---|
| Minimum | 0.00 | 0.00 | 0.01 | 0.00 |
| Maximum | 0.16 | 0.50 | 0.50 | 0.38 |
| Mean | 0.08 | 0.27 | 0.20 | 0.22 |

Ho = observed heterozygosity, He = expected heterozygosity, MAF = minor allele frequency, PIC = polymorphic information content.

consisted of seven genotypes obtained from USDA and IITA. The phylogenetic tree also showed three sub-populations with higher degrees of admixture similar to the BIC (Fig 5).

Similarly, the DAPC assigned the genotypes to three cluster groups and clearly showed a higher degree of admixture among the genotypes. The first and second PCs accounted for 35.8 and 13.8% of the total variation, respectively. The members of Cluster III were the most compact in distribution, while those of Cluster I were the most widely distributed along with the axes of the first two PCs (Fig 6).

## Genetic distance between genotypes based on 10,630 SNP markers

The genetic distances across the genotypes are summarized in **S4** Table. In cluster one, the highest genetic distance (0.335) was observed between genotypes SY007 and SY024, while the lowest genetic distance (0.14) was observed between genotypes SY013 and SY012. In cluster two, the maximum genetic distance (0.335) was observed between genotypes SY026 and SY030, while the minimum genetic distance (0.012) was observed between genotypes SY037 and

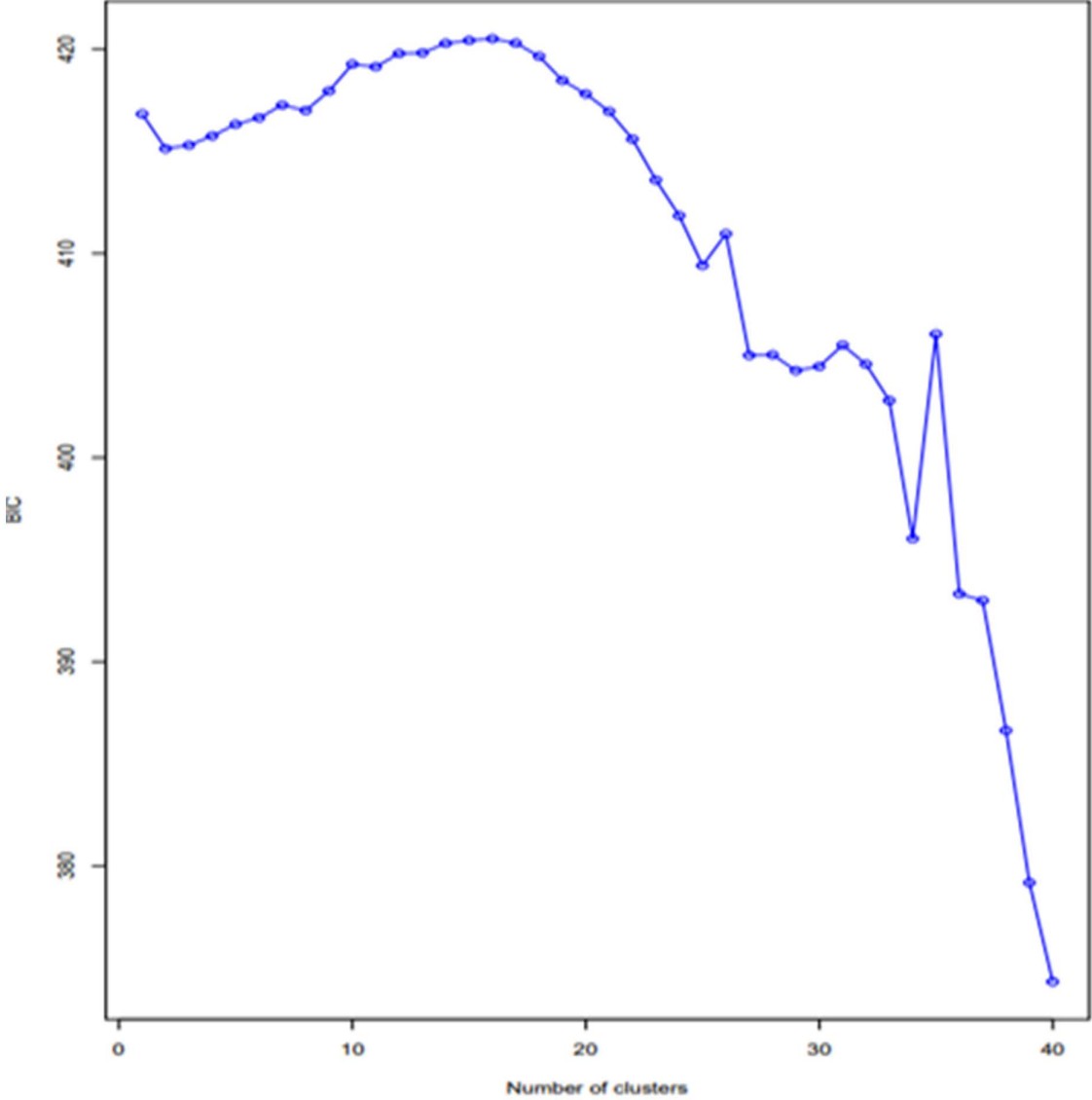

**Fig 3. Graph showing the number of clusters vs the Bayesian Information Criterion (BIC).**

SY029. In cluster three, the widest genetic distance (0.335) was observed between genotypes SY030 and SY026, while the shortest genetic distance (0.014) was observed between genotypes SY026 and SY020.

## Discussion

Genetic diversity analysis in crops is required for the success of any plant breeding effort [58]. Hence, evaluating crops' population structure and genetic diversity is vital for adopting effective genetic resource management and conservation measures [59]. The genotype × location interactions across the three locations were significant for grain yield, and highly significant for days to 95% maturity, and days to 50% flowering, implying genotypes exhibited varying relative performances across locations for these traits. The results are in line with the findings of Tesfaye et al. [60], Krisnawati and Adie [61], and Pedro [62], who reported a highly significant genotype x environment interaction for days to 50%

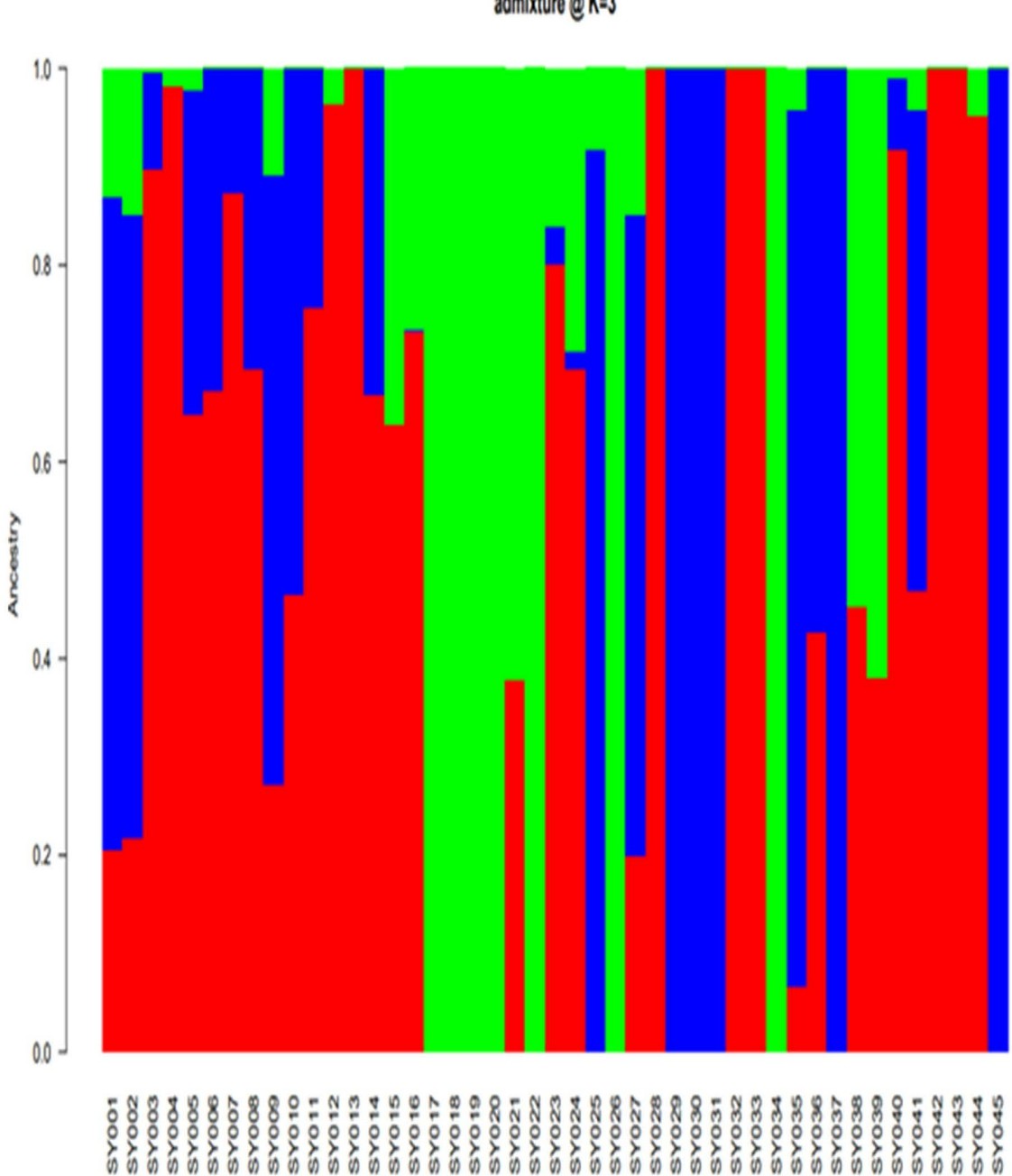

**Fig 4. Population structure of the studied soybean genotypes based on the SNP markers.** Red, blue, and green color denotes sub-populations I, II, and III, respectively.

flowering, days to 95% maturity, and grain yield. Njoroge et al. [63], Nachilima [64], and Abebe et al. [65] also reported significant genotype x environment interaction for grain yield. Five genotypes, i.e., G02, G10, G11, G01, and G24 showed significantly superior performance for grain yield (Table 5). These genotypes can be released as varieties for direct production by farmers after further evaluation or can be used as parents in the soybean hybridization programs.

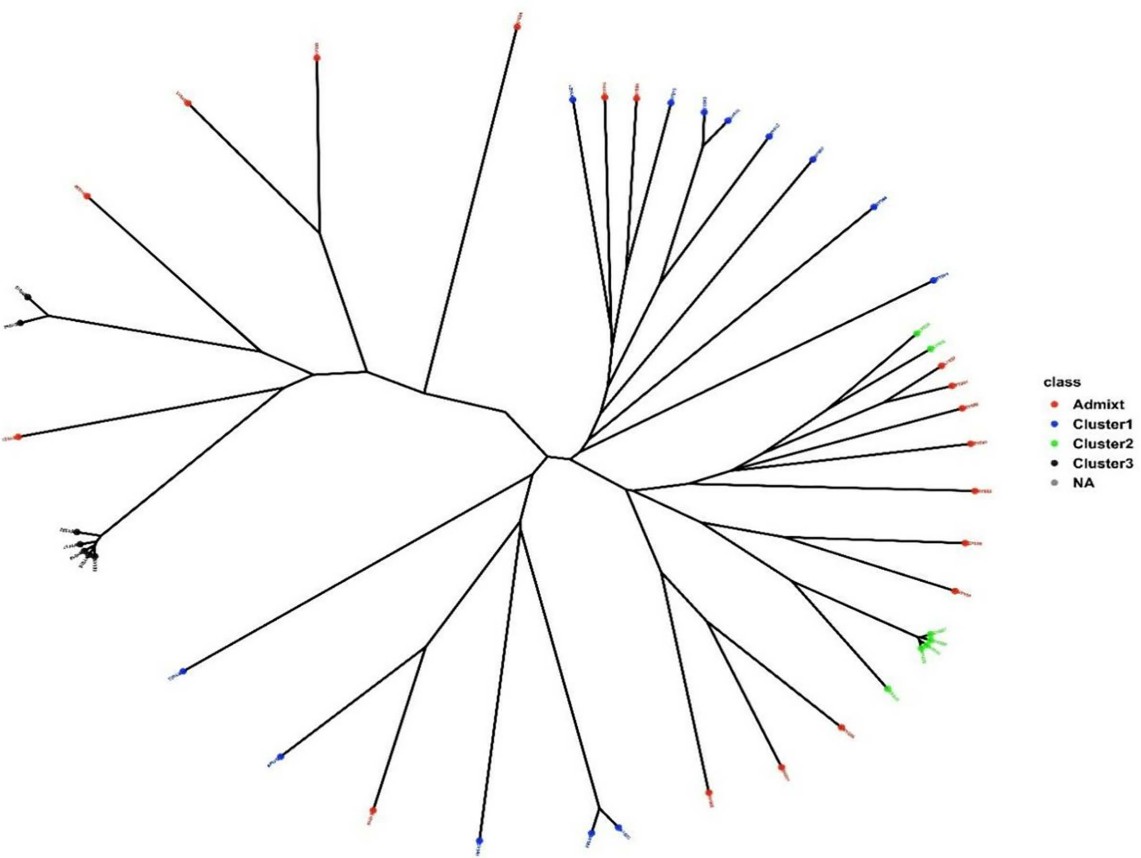

**Fig 5. The phylogenetic trees of the studied soybean genotypes based on the SNP markers.** Dots indicating individual genotypes.

Getnet [66], Beyene et al. [67], and Thio et al. [68] also obtained different mean yield performances among soybean genotypes in their studies.

Days to 95% maturity varied from 112 to 121 days, with G32 and G33 maturing earlier than the rest, while G25 maturing relatively late. Beyene et al. [67], Thio et al. [68], Sileshi [69], Goonde and Ayana [70] and Yirga et al. [71] reported comparable trends in variability in days to maturity for different soybean genotypes. Days to 50% flowering varied from 44 to 55 days with the earliest genotype being G23, followed by G33. The genotypes that took longer to flower were G04, G05, G08, and G13 all with 55 days. The results for days to 50% flowering agree with the findings of Goonde and Ayana [70] and Jandong et al. [72], and Akter [73]. Shattering ranged from 0.96 to 3.5 with the highest tolerant genotypes being G43, G07, and G27, whereas G45 was highly susceptible to pod-shattering. This result aligns with Fatima et al. [45] and Aondover et al. [74], who reported different mean performances in pod shattering. G44 recorded the maximum lodging score, while the minimum was observed in G22. This aligns with the work of Antwi-Boasiako [75], who found significant variation in tolerance to lodging and shattering across the 34 soybean genotypes evaluated. The root nodule score ranged from 2.17 to 3.61 with an average of 3.13. Among the 45 soybean genotypes, 27 nodulated more efficiently, and the best genotypes were G02, G16, G24, G29, G45, G01, G07, and G15. These results agree with those of Thio et al. [68] and Bello et al. [76], who reported different mean performances in root nodule scores. The plant height among the genotypes varied from 63.6 to 102 cm. The tallest plant height was recorded for G17, whereas the shortest was recorded on G02, which was also the top-performing genotype for grain yield. The results of this study is in line with the findings of Thio

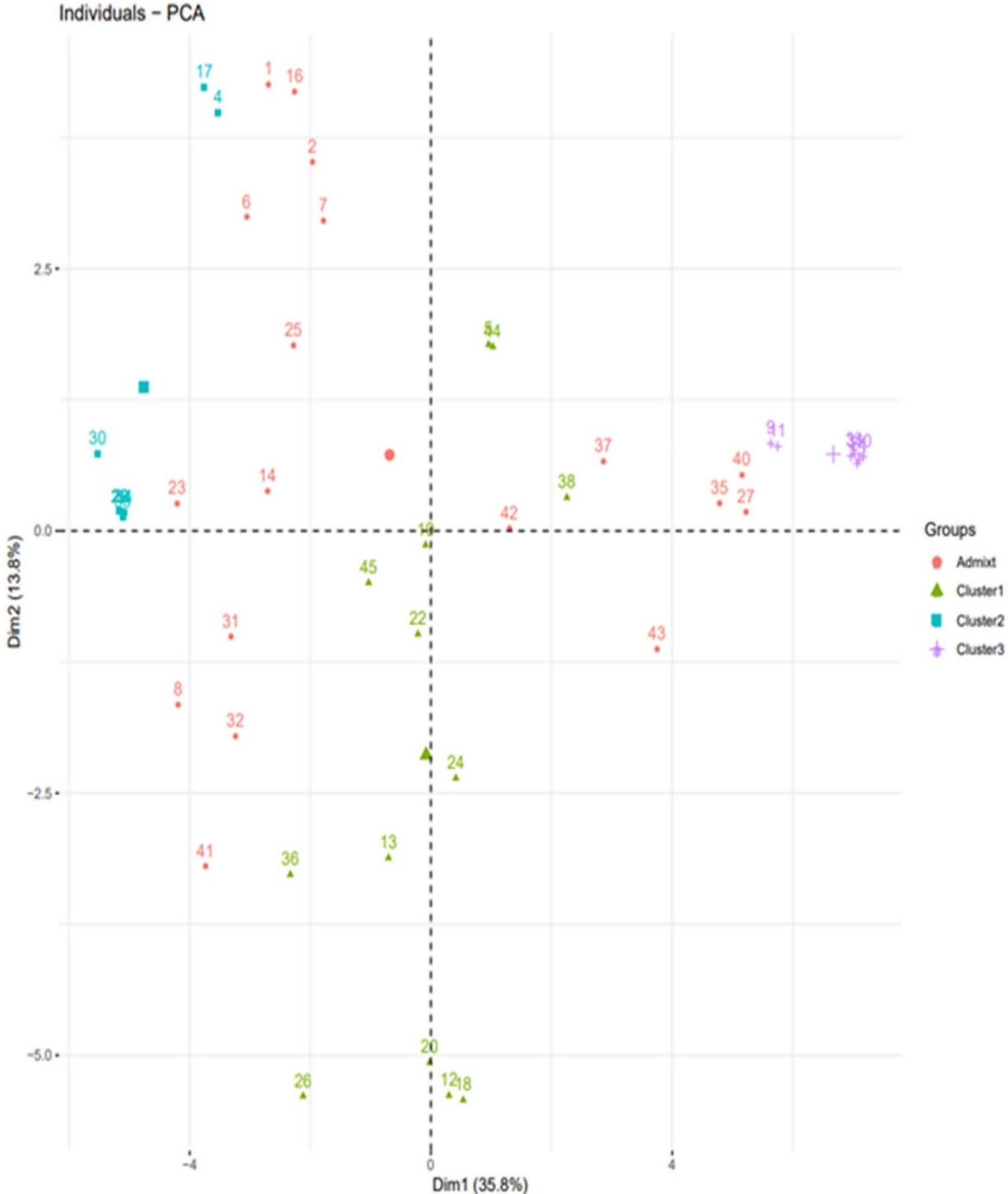

**Fig 6. Scatter plot based on the first two PCs obtained from DAPC based on the SNP markers displaying the distribution of the studied genotypes.** Note Each color corresponds to population structuring and grouping.

et al. [68] and Njoroge and Njeru [77], who reported plant height ranged from 51.1 cm to 102.8 cm and 60 cm to 109 cm, respectively. In contrast, Hizli et al. [78] reported up to 151.8 cm plant height, which is taller than the heights found in this study.

Principal component analysis (PCA) aids in simplifying multidimensional data by breaking down its complexity into simpler principal components (PCs) by retaining important information [79,80]. The first three PCs with eigenvalues greater than one, which accounted for about 64.8% of the total variation were considered important (Table 5). In a similar study on soybeans, Aondover et al. [74] reported that the first three principal components accounted for 76.4% of the total variation. Similarly, Verma et al. [81] reported that the first three PCs contributed 91.7% of the total variation. In their study, Akter [73] and Vijayakumar et al. [80] also reported that the first three PCs accounted for 77.7% and 73.7% of the total variation, respectively. Each PC is independently associated with the different yield and yield-related variables examined. The first PC that showed the highest contributions (27.2%) to the total variation was highly and positively associated with lodging and highly and negatively associated with days to 95% maturity, indicating the importance of these traits in the genetic improvement of the studied germplasm. The average contributions of RtNd, HSW, PH, D50F, and Lodg were high in the principal axes. This finding implies that these traits account for the majority of the variations in soybeans and further influence the yield of soybeans. This finding aligns with the results of Aondover et al. [74], Adetiloye et al. [82], Sivabharathi et al. [83], Vianna et al. [84], and Leite et al. [85], who reported the mean contributions of RtNd, HSW, PH, D50F, and Lodg were high in the principal axes. Thus, these traits must be considered when selecting grain yield. Soybean genotypes were spread over the biplot based on the first two PCs, also reflecting high variability in the studied germplasm. Genotypes G15, G24, G45, G13, and G02 were far from the origin and identified as highly interactive and diverse genotypes; whereas G23, G43, G11, G07, G01, and G09 were located close to the origin and considered stable genotypes or less diverse genotypes (Fig 1). Comparable results were reported by Sivabharathi et al. [83], who performed principal component analysis on 135 soybean genotypes and reported genotypes MACS 1460, EC 18736, and PK 1038 as highly divergent genotypes, while JS 89–24, NRC 25, NRC 2007-G-1–13, NRC 43 and PK 7247 genotypes were less diverse and stable.

The studied genotypes were grouped into five distinct clusters, which also indicates the divergent nature of these germplasm. This result was comparable to the findings of Getnet [66] and Ghiday and Sentayehu [86], who reported the clustering of 49 soybean genotypes into five and three distinct clusters, respectively. Similarly, Yirga et al. [71] performed cluster analysis of 100 soybean genotypes based on morphological traits that were grouped into five different clusters, which implied the presence of genetic variability among the tested genotypes. Adetiloye et al. [82] also reported the clustering of 43 accessions into nine distinct clusters. These findings indicate the studied soybean genotypes exhibited wide genetic diversity. Utilizing genetically diverse parents in hybridization programs is critical in crop improvement activities [73] in ensuring high genetic recombinations in the progeny populations. Considering the cluster mean values presented in Table 6 and the significant differences revealed by F-test, the observed significant differences among clusters for key agronomic traits highlight the potential of cluster-based selection in soybean breeding. Notably, Cluster IV, which exhibited the highest grain yield, and Cluster V, which combined the latest maturity duration and tolerance to lodging, represent valuable genetic sources for breeding programs. For instance, Cluster II, with its early maturity and shorter plant height, may be suitable for regions with shorter growing periods. The present outcome supports prior findings [87–90].

In this study, the observed heterozygosity (Ho) of 0.08 was lower than the expected heterozygosity of the genotypes, possibly due to soybeans' high self-pollinating nature [91]. This implies high possibilities of inbreeding and fixation at most loci [92,93]. The mean observed heterozygosity of 0.08 obtained in this study is higher compared to the 0.02 and 0.06 reported by Tsindi et al. [35] and Chiemeke et al. [94], respectively, in soybeans using SNP markers. However, it is lower compared to the 0.22 stated by Yohane et al. [95] when assessing 81 pigeon pea genotypes based on 4122 SNP markers, but similar to the results of Gumede et al. [59], when assessing 90 cowpea accessions using 5864 SNP markers. The average expected heterozygosity (He = 0.27) found in the current study was lower than the He of 0.31 reported by Abebe

et al. [34] and Tsindi et al. [35] evaluating 65 and 210 soybean genotypes using 1223 and 403 SNP markers, respectively. The minor allele frequency (MAF) values indicate the prevalence of the less common allele at a locus, helping to distinguish common from rare variants. Minor allele frequency (MAF) varied from 0.01 to 0.50 with an average of 0.20. The average MAF value found in this study was greater than the 0.24 reported by Tsindi et al. [35] based on the 403 SNPs in 210 soybean genotypes, 0.25 reported by Naflath et al. [96] based on the 29,955 SNPs in 96 soybean genotypes and 0.32 reported by Chander et al. [97] based on 186 SNPs in 155 soybean lines. Dube et al. [40] found similar results in maize with the MAF ranging from 0.01 to 0.5. A mean PIC value of 0.22 indicates that the markers had moderate informativeness for distinguishing among genotypes. A mean PIC value of 0.22 in the current study was lower than the PIC values reported in various crops such as 0.25 in soybean [34], 0.31 in maize [40], 0.36 in wing yam [41], 0.29 in wheat [98], 0.27 in cowpea [59], 0.26 in pea [99]. However, it is comparable to the results in other legumes, i.e., 0.22 in common bean [100] and 0.23 in cowpea [101].

The observed (Ho = 0.08) and expected (He = 0.27) heterozygosity values in this study reflect soybean's self-pollinating nature, which naturally results in lower heterozygosity [91,102]. However, the level of expected heterozygosity still indicates the presence of useful genetic variation across the population [103]. This moderate diversity is essential for breeders because it enables effective selection of genotypes with favorable traits [59,104]. The disparity between Ho and He also suggests some degree of inbreeding and allele fixation, highlighting the need for incorporating genetically diverse parents in the crossing program to maintain and further increase the genetic base of IITA soybean [92,97,104]. Besides, by comparing Ho and He to other soybean germplasm such as Abebe et al. [34], Tsindi et al. [35], and Chiemeke et al. [94], breeders can pinpoint the most diverse parents for hybridization and thus maximize recombination of useful alleles [105,106]. Additionally, moderate average values PIC (0.22) confirm that the SNP markers used are informative enough to distinguish genotypes and support population structure analysis [54,97]. These diversity indices together provide valuable guidance for parental selection, germplasm conservation, and marker deployment in soybean improvement programs [54,105].

Population structure analysis is the process of determining a breeding line's ancestry using genotypic data [38,107]. The SNP analysis identified three sub-populations (K = 3) among the 45 soybean genotypes based on the optimal K = 3 determined according to the BIC method, which was not consistent with the results of phenotypic clustering. The inconsistency might be that most of the phenotypic traits are controlled by polygenes, and environmental conditions highly influence these traits [108,109]. In the same way, the phylogenetic tree and DAPC also showed three sub-populations with higher degrees of admixture. The high degree of admixture found suggests that there was either gene flow or that these subpopulations had a common ancestor [91]. Chander et al. [97] found comparable amounts of admixture in their study of 155 soybean genotypes, predominantly IITA-bred soybean varieties. The overall results of the population structure analysis are in agreement with the results of Abebe et al. [34] and Fatokun et al. [101], who identified three sub-populations among the 65 soybean genotypes and 298 cowpea accessions, respectively based on BIC and DAPC methods.

The analysis of genetic distances within clusters revealed considerable variation, reflecting the underlying genetic diversity among the genotypes. In cluster I, the greatest genetic distance was recorded between genotypes SY007 and SY024, suggesting substantial genetic differentiation between these two individuals. This could imply limited recent shared ancestry or contrasting selection pressures. On the other hand, the lowest genetic distance was observed between SY013 and SY012, indicating a closer genetic relationship, possibly due to recent common ancestry or shared breeding lineage [107,110]. In cluster II, the widest genetic distance was found between genotypes SY026 and SY030, mirroring the divergence seen in Cluster 1. Such high differentiation might suggest that these genotypes belong to distinct subpopulations or have undergone different evolutionary pressures. Conversely, the minimal genetic distance in this cluster was surprisingly low between SY037 and SY029, indicating a near-identical genetic makeup. This could reflect either clonal propagation, recent hybridization, or strong genetic conservation between the two genotypes. Cluster III also exhibited a maximum genetic distance, again between SY030 and SY026, reinforcing the notion of pronounced genetic divergence between

these two genotypes across clusters. The lowest genetic distance in this cluster was between SY026 and SY020, suggesting a close relationship and potential kinship or shared origin. This pattern of high and low pairwise distances within clusters indicates that while clustering helps group genetically similar individuals, considerable diversity still exists within each group, which is critical for maintaining adaptive potential and guiding breeding decisions. Various genetic distances within soybean germplasm have been observed in a similar study by [35,37,102,111]

## Conclusion

The combined analysis of variance revealed significant differences among genotypes, locations, and the genotype×location interaction for days to 95% maturity and grain yield, indicating that genotypes exhibited different relative performances across locations for these traits. The genotypes G02, G10, G11, G01, and G24 were found to be the most high-yielding (3050−2990 kg/ha). The first three principal components explained 64.8% of the variation, with traits like HSW, RtNd, PH, Lodg, and D50F contributing the most in the principal axes. Cluster analysis grouped the 45 genotypes into five clusters, suggesting moderate variation. Based on cluster mean values and the significant differences, genotypes selected from distant clusters such as clusters II and IV, were desirable for use as parents in future hybridization programs to develop high-yielding soybean varieties.

From the molecular analysis, 10,630 SNP markers were used, showing moderate informativeness with a mean polymorphic information content (PIC) of 0.22. The mean observed heterozygosity (Ho = 0.08), expected heterozygosity (He = 0.27), and minor allele frequency (MAF = 0.20) values indicate moderate genetic variability within the genotypes. Population structure analysis using DAPC and BIC identified three subpopulations, confirming considerable genetic diversity among the studied soybean genotypes.

## Supporting information

**S1 Table. Eigenvalues, proportion of variance, and cumulative proportion of the studied soybean genotypes for eight agronomic and yield traits.** D50F = days to 50% flowering, SHS = shattering score, RtNd = root nodule score, Lodg = lodging score, PH = plant height (cm), D95M = days to 95% maturity, HSW = hundred seed weight (gm), GY = grain yield (kg/ha).
(CSV)

**S2 Table. List of genotypes and their grouping based on 10,630 SNP markers using the 70% cut-off criterion of the membership probability threshold.**
(CSV)

**S3 Table. Mean performances of 45 soybean genotypes evaluated at three locations in Nigeria for agronomic and yield-related traits.** RtNd = root nodule score, Lodg = lodging score, D50F = days to 50% flowering, PH = plant height in centimeter, D95M = days to 95% maturity, SHS = shattering score, GY = grain yield in kilogram per hectare, SE = standard error, LSD = least significant difference.
(DOCX)

**S4 Table. Genetic distance matrix of the 45 soybean genotypes based on SNP markers.**
(CSV)

**S5 Table. Location-wise mean performances for grain yield, D50F, and D95M of the study genotypes.**
(XLSX)

**S1 Fig. Summary statistics of 10,630 single nucleotide polymorphism (SNP) markers used for genotyping 45 soybean genotypes.**
(PNG)

## Acknowledgments

The first author is grateful for the scholarship granted by the African Union Commission for her MSc studies at the Pan African University Life and Earth Sciences Institute (including Health and Agriculture), University of Ibadan, Nigeria. We would like to acknowledge the staff and technicians of the soybean breeding and biometric unit, at the International Institute of Tropical Agriculture for all their invaluable guidance.

## Author contributions

**Conceptualization:** Abebawork Tilahun Assfaw, Olasanmi Bunmi, Abush Tesfaye Abebe.

**Data curation:** Abebawork Tilahun Assfaw.

**Formal analysis:** Abebawork Tilahun Assfaw, Agre Paterne, Kayode Fowobaje.

**Funding acquisition:** Abebawork Tilahun Assfaw, Godfree Chigeza, Hapson Mushoriwa, Abush Tesfaye Abebe.

**Investigation:** Abebawork Tilahun Assfaw.

**Methodology:** Abebawork Tilahun Assfaw, Agre Paterne, Kayode Fowobaje, Abush Tesfaye Abebe.

**Project administration:** Abebawork Tilahun Assfaw, Godfree Chigeza, Hapson Mushoriwa, Abush Tesfaye Abebe.

**Resources:** Abebawork Tilahun Assfaw, Abush Tesfaye Abebe.

**Software:** Abebawork Tilahun Assfaw, Agre Paterne, Kayode Fowobaje.

**Supervision:** Abebawork Tilahun Assfaw, Olasanmi Bunmi, Abush Tesfaye Abebe.

**Validation:** Abebawork Tilahun Assfaw, Agre Paterne, Abush Tesfaye Abebe.

**Visualization:** Abebawork Tilahun Assfaw, Agre Paterne, Kayode Fowobaje.

**Writing – original draft:** Abebawork Tilahun Assfaw.

**Writing – review & editing:** Abebawork Tilahun Assfaw, Olasanmi Bunmi, Abush Tesfaye Abebe.

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
