## [Decision Letter · Decision Letter 0]

11 Apr 2025

Dear Dr. Assfaw,

Thank you for submitting your manuscript to PLOS ONE. After careful consideration, we feel that it has merit but does not fully meet PLOS ONE’s publication criteria as it currently stands. Therefore, we invite you to submit a revised version of the manuscript that addresses the points raised during the review process.

We look forward to receiving your revised manuscript.

Kind regards,

Bahram Heidari

Academic Editor

PLOS ONE

Additional Editor Comments (if provided):

Reviewers' comments:

Reviewer's Responses to Questions

**Comments to the Author**

1. Is the manuscript technically sound, and do the data support the conclusions?

Reviewer #1: Yes

Reviewer #2: Partly

2. Has the statistical analysis been performed appropriately and rigorously?

Reviewer #1: No

Reviewer #2: Yes

3. Have the authors made all data underlying the findings in their manuscript fully available?

Reviewer #1: No

Reviewer #2: Yes

4. Is the manuscript presented in an intelligible fashion and written in standard English?

Reviewer #1: Yes

Reviewer #2: Yes

Reviewer #1: This study examines the genetic diversity and population structure of 45 soybean genotypes from the IITA breeding program, utilizing both phenotypic traits and SNP markers. The authors employ multivariate statistical methods such as principal component analysis (PCA) and cluster analysis to assess the genotypic and phenotypic variation. However, there are a few suggestions that could further improve the manuscript, as outlined below:

Line 39: Replace this "Soybean [Glycine max (L.) Merrill], is the leading oil seed crop, accounting for 57% of the global oil production." With this "Soybean [Glycine max (L.) Merrill] is one of the leading oilseed crops globally, accounting for approximately 57% of global vegetable oil production."

Line 41-43: Replace this "It is cultivated in many countries and is a major source of vegetable oil and protein used in food, animal feed, and for industrial purposes." With this "It is cultivated in numerous countries and serves as a major source of vegetable oil and protein, essential in food, feed, and various industrial applications."

Line 44-45: Replace this "It was domesticated in the eleventh (11th) century BC around Northeast China and was then transported throughout Asia, the USA, Brazil, and Argentina." With this "Soybeans were domesticated around the 11th century BC in Northeast China and subsequently spread across Asia, the USA, Brazil, and Argentina."

Line 46-47: Replace this "Nigeria is the second-largest producer of soybeans in Africa, after South Africa. The first soybeans were brought into Nigeria in 1908, but the first successful cultivation was in 1937 with the Malayan variety, which was reported as commercial production in Benue State." With this "Nigeria ranks as the second-largest producer of soybeans in Africa, following South Africa. Soybeans were first introduced to Nigeria in 1908, with successful commercial cultivation beginning in 1937, using the Malayan variety in Benue State."

Line 49-50: Replace this "The crop is adapted to various environmental conditions and is primarily cultivated under rain-fed conditions." With this "The crop is adapted to diverse environmental conditions and is predominantly grown under rain-fed conditions."

Lines 55–57: How can the genetic improvement of soybeans be specifically tailored to address malnutrition in Nigeria or similar regions? Are there particular traits being targeted that would make soybeans more beneficial nutritionally?

Line 184: Have you considered including an Analysis of Variance (ANOVA) along with mean comparison for the phenotypic traits assessed before performing the multivariate statistical analyses such as PCA and clustering? This could help to highlight significant phenotypic variation among the genotypes and provide a clearer basis for the subsequent analyses.

Line 259: The discussion section could benefit from incorporating the results from the ANOVA and mean comparison analyses, especially when interpreting the genetic variability observed in the study.

Reviewer #2: 1- The results of ANOVA are needed.

2- Add the number of plants used for phenotypical measurements.

3- Add the method of cluster analysis.

4- Write the formulas used for parameters estimation to the materials and methods section.

5- Consider eigenvectors equal to or greater than 0.4 as high in the PCA analysis and reinterpret.

6- Calculation of genetic distances between cluster groups and genetoypes are required.

7- I have mentioned some suggestions in the text of the article that should be corrected.

**Do you want your identity to be public for this peer review?** For information about this choice, including consent withdrawal, please see our Privacy Policy

Reviewer #1: No

Reviewer #2: **Yes: ** Massoud Dehdari

---

## [Author Response · Author response to Decision Letter 1]

2 Jun 2025

Response: Thank you for the valuable comment. Accordingly, we have added the ANOVA table and mean comparison results for the phenotypic traits (see new Table 4 and S3 Table in the Results section). Although we did not conduct testing on the data for homogeneity and normality before the ANOVA analysis because the fitted linear mixed-effect model (using the lmerTest package in R) is robust to violations of both homogeneity of variance and normality. This model can explicitly handle heteroscedasticity through the inclusion of random effects and effectively handles deviations from normality, thus ensuring the validity and reliability of our results without requiring separate assumption testing.

Response: The sentence has been revised accordingly in the introduction.

---

## [Decision Letter · Decision Letter 1]

16 Jun 2025

Dear Dr. Assfaw,

Thank you for submitting your manuscript to PLOS ONE. After careful consideration, we feel that it has merit but does not fully meet PLOS ONE’s publication criteria as it currently stands. Therefore, we invite you to submit a revised version of the manuscript that addresses the points raised during the review process.

We look forward to receiving your revised manuscript.

Kind regards,

Yêyinou Laura Estelle Loko

Academic Editor

PLOS ONE

Additional Editor Comments:

Reviewer 1

The authors reported on “Genetic diversity and population structure analysis of soybean [Glycine max (L.) Merrill] genotypes based on agro-morphological traits and SNP markers”, an aspect that is key to the success of every breeding program as a perquisite to successful varietal development. The authors have done well in the presentation of the manuscript sections though with minor corrections to improve the manuscript quality. My few comments for each section are below.

Abstract

L24&235: “…genotypes were 25 further assessed for SNPs-based genotypic diversity analyses”. Please include the type of marker and the quantity.

L28: take out the word “corresponding” the markers are not corresponding to the traits used for pheno-based phylogeny

L29 and 30: it is better to list a few of the desirable traits in the genotypes rather than just using a general perception.

L33: you said “considerable genetic variation” but concluded with “high genetic variation” which one should the reader accept?

Introduction

L53: I think you meant “…..high-yielding, climate, and stress-resilient…..”

Materials and Methods

L115 – 122: Experimental materials should come before experimental design and management.

L130: should be “…agro-morphological traits including plant height…”

L134 – 145: I assume you did ANOVA but chose not to report it. If no please it should be part of your results or at least in your supplementary, file but reported in your main text. ANOVA is the basis for establishing variability that is your diversity per se. you have data from three locations, readers will like to see environment, genotype, and their interaction effects prior to be interested in other results. Please include the process in handling these aspects of the analysis in the phenotypic data analysis focusing on choice of model used, justification, and how you generated the data for cluster analysis. You did cluster analysis but you kept information on the data used (BLUES or BLUP) and why?

L135: what do you mean by similarity coefficient? Did you use similarity matrix or dissimilarity matrix for clustering? In my opinion, your focus is diversity so dissimilarity matrix like, Gower and Euclidean are fine.

L143: You did PCA with two packages. I am familiar with both packages and can tell that both of them do not do PCA. Only one does the PCA while the other is for visualization. So please be specific which one was used for PCA and which was for visualization.

L149: what do you mean immature? I assume you meant fully expanded but young leaf. If so, please correct and if not please clarify.

L170 – 183: your MAF threshold seems very low (0.01), the standard is usually 0.05. Do you have a reason for bringing it down to 0.01? how did you handle markers with high heterozygosity? Did you check for call rate prior to QC and what was your threshold?

L177 and 178: You used both VCFtool and PLINK for filtering and marker diversity indices assessment. Please be clear which one is for which activity? BIC is not to estimate but to define optimum subpopulation.

Results

NB: Please include the proposed analysis in the materials and methods in the results section.

L185 – 196: You cited table 4 before fig 1. Please arrange the results in same manner.

L210: fig 2 should be placed after the text just before table 5.

L212 – 215: there is grammar issue here, please attend to it.

L220: Please subject the cluster performance to f-statistics to see whether the cluster groups are significantly different from one another for each of the traits. That way, you can identify which clusters actually possess what traits with statistical confidence.

L237: Your BIC vs cluster number graph had a sharp drop from one to two and at this point was also the lowest indicating that two was the optimal subpopulation but you said three. Could you justify? I suggest you verify the subpopulation with alternative analysis like PCA scree plot. If it happens that you consider two subpopulations then DAPC may have to change.

L249: Fig 5 seems to have been misplaced. The figure was titled phylogenetic three but was placed under population structure. If I am wrong then please provide the results for the phylogenetic analysis. This can be compared to the phylogenetic analysis from the phenotypic data to see how the genotypes grouping aligns.

Discussion

L313: The authors said “The minor allele frequency (MAF) values measure the ability of markers to discriminate among genotypes” this is new and contrary to what is known about MAF. If MAF does this, then what is function of PIC?

Conclusion

The conclusion is too long and needs to be reduced and more concise.

Other information

The figures not easily readable.

Reviewer 2

1-Add error1 and error2 as SOV in the ANOVA table

2-For yield, D50F and D95M traits, you should introduce the superior genotype at each location because the interaction has become significant.

Reviewers' comments:

Reviewer's Responses to Questions

**Comments to the Author**

Reviewer #1: All comments have been addressed

Reviewer #2: All comments have been addressed

Reviewer #3: (No Response)

2. Is the manuscript technically sound, and do the data support the conclusions?

Reviewer #1: (No Response)

Reviewer #2: Yes

Reviewer #3: Yes

3. Has the statistical analysis been performed appropriately and rigorously?

Reviewer #1: Yes

Reviewer #2: Yes

Reviewer #3: Yes

4. Have the authors made all data underlying the findings in their manuscript fully available?

Reviewer #1: Yes

Reviewer #2: Yes

Reviewer #3: Yes

5. Is the manuscript presented in an intelligible fashion and written in standard English?

Reviewer #1: Yes

Reviewer #2: (No Response)

Reviewer #3: Yes

Reviewer #1: (No Response)

Reviewer #2: 1-Add error1 and error2 as SOV in the ANOVA table

2-For yield, D50F and D95M traits, you should introduce the superior genotype at each location because the interaction has become significant.

Reviewer #3: The authors reported on “Genetic diversity and population structure analysis of soybean [Glycine max (L.) Merrill] genotypes based on agro-morphological traits and SNP markers”, an aspect that is key to the success of every breeding program as a perquisite to successful varietal development. The authors have done well in the presentation of the manuscript sections though with minor corrections to improve the manuscript quality. My few comments for each section are below.

Abstract

L24&235: “…genotypes were 25 further assessed for SNPs-based genotypic diversity analyses”. Please include the type of marker and the quantity.

L28: take out the word “corresponding” the markers are not corresponding to the traits used for pheno-based phylogeny

L29 and 30: it is better to list a few of the desirable traits in the genotypes rather than just using a general perception.

L33: you said “considerable genetic variation” but concluded with “high genetic variation” which one should the reader accept?

Introduction

L53: I think you meant “…..high-yielding, climate, and stress-resilient…..”

Materials and Methods

L115 – 122: Experimental materials should come before experimental design and management.

L130: should be “…agro-morphological traits including plant height…”

L134 – 145: I assume you did ANOVA but chose not to report it. If no please it should be part of your results or at least in your supplementary, file but reported in your main text. ANOVA is the basis for establishing variability that is your diversity per se. you have data from three locations, readers will like to see environment, genotype, and their interaction effects prior to be interested in other results. Please include the process in handling these aspects of the analysis in the phenotypic data analysis focusing on choice of model used, justification, and how you generated the data for cluster analysis. You did cluster analysis but you kept information on the data used (BLUES or BLUP) and why?

L135: what do you mean by similarity coefficient? Did you use similarity matrix or dissimilarity matrix for clustering? In my opinion, your focus is diversity so dissimilarity matrix like, Gower and Euclidean are fine.

L143: You did PCA with two packages. I am familiar with both packages and can tell that both of them do not do PCA. Only one does the PCA while the other is for visualization. So please be specific which one was used for PCA and which was for visualization.

L149: what do you mean immature? I assume you meant fully expanded but young leaf. If so, please correct and if not please clarify.

L170 – 183: your MAF threshold seems very low (0.01), the standard is usually 0.05. Do you have a reason for bringing it down to 0.01? how did you handle markers with high heterozygosity? Did you check for call rate prior to QC and what was your threshold?

L177 and 178: You used both VCFtool and PLINK for filtering and marker diversity indices assessment. Please be clear which one is for which activity? BIC is not to estimate but to define optimum subpopulation.

Results

NB: Please include the proposed analysis in the materials and methods in the results section.

L185 – 196: You cited table 4 before fig 1. Please arrange the results in same manner.

L210: fig 2 should be placed after the text just before table 5.

L212 – 215: there is grammar issue here, please attend to it.

L220: Please subject the cluster performance to f-statistics to see whether the cluster groups are significantly different from one another for each of the traits. That way, you can identify which clusters actually possess what traits with statistical confidence.

L237: Your BIC vs cluster number graph had a sharp drop from one to two and at this point was also the lowest indicating that two was the optimal subpopulation but you said three. Could you justify? I suggest you verify the subpopulation with alternative analysis like PCA scree plot. If it happens that you consider two subpopulations then DAPC may have to change.

L249: Fig 5 seems to have been misplaced. The figure was titled phylogenetic three but was placed under population structure. If I am wrong then please provide the results for the phylogenetic analysis. This can be compared to the phylogenetic analysis from the phenotypic data to see how the genotypes grouping aligns.

Discussion

L313: The authors said “The minor allele frequency (MAF) values measure the ability of markers to discriminate among genotypes” this is new and contrary to what is known about MAF. If MAF does this, then what is function of PIC?

Conclusion

The conclusion is too long and needs to be reduced and more concise.

Other information

The figures not easily readable.

**Do you want your identity to be public for this peer review?** For information about this choice, including consent withdrawal, please see our Privacy Policy

Reviewer #1: **Yes: ** Nikwan Shariatipour

Reviewer #2: **Yes: ** Massoud Dehdari

Reviewer #3: **Yes: ** DR. ADEJUMOBI IDRIS ISHOLA

---

## [Author Response · Author response to Decision Letter 2]

30 Jul 2025

Reviewer #1

L24&235: “…genotypes were 25 further assessed for SNPs-based genotypic diversity analyses”. Please include the type of marker and the quantity.

Response: Revised to read: “…the genotypes were further assessed using 10,630 SNP markers obtained from DArTseq genotyping.

L28: take out the word “corresponding” the markers are not corresponding to the traits used for pheno-based phylogeny

Response: The word “corresponding” has been removed for clarity.

L29 and 30: it is better to list a few of the desirable traits in the genotypes rather than just using a general perception.

Response: Thank you for the suggestion. We have revised the sentence in the abstract to include specific desirable traits. The updated sentence now highlights traits such as grain yield, days to 50% flowering and days to 95% maturity, hundred seed weight, lodging, plant height, and nodulation, which were key in differentiating the genotypes.

---

## [Decision Letter · Decision Letter 2]

20 Aug 2025

Dear Dr. Assfaw,

Thank you for submitting your manuscript to PLOS ONE. After careful consideration, we feel that it has merit but does not fully meet PLOS ONE’s publication criteria as it currently stands. Therefore, we invite you to submit a revised version of the manuscript that addresses the points raised during the review process.

We look forward to receiving your revised manuscript.

Kind regards,

Yêyinou Laura Estelle Loko

Academic Editor

PLOS ONE

Journal Requirements:

Reviewers' comments:

Reviewer's Responses to Questions

**Comments to the Author**

Reviewer #2: All comments have been addressed

Reviewer #3: All comments have been addressed

2. Is the manuscript technically sound, and do the data support the conclusions?

Reviewer #2: Yes

Reviewer #3: Yes

3. Has the statistical analysis been performed appropriately and rigorously?

Reviewer #2: Yes

Reviewer #3: Yes

4. Have the authors made all data underlying the findings in their manuscript fully available?

Reviewer #2: Yes

Reviewer #3: Yes

5. Is the manuscript presented in an intelligible fashion and written in standard English?

Reviewer #2: Yes

Reviewer #3: Yes

Reviewer #2: 1-DF=g(r-1)=45*(3-1)=90

I think the DF for Error2 is 90. Please response to this comment.

2-For yield, D50F and D95M traits, you should introduce the superior genotype at each location because the interaction has become significant.

Reviewer #3: Table 2: what does the SNP designation in the Table represent?

Table 3: the caption should be well-placed on top of the Table and not in the first cell. In addition, add the status of each trait as separate column in the Table (qualitative or quantitative).

Line 146: linear mixed model is LMM or call it mixed linear model (MLM) not LMER (this is a function in lmerTest R-package).

Line 157: I do not think LSD can do a good mean comparison for 45 genotypes if you look thoroughly on how it calculates the critical value for comparison. Why not explore HSD that is more robust.

Line 160 – 165: Kindly include the number of traits used for the cluster analysis and how they were selected. The method is Ward.D2 and not Ward.D2 . cluster analysis is performed using based R function not dendextend package. The package is only used to plot the dendrogram.

Line 168-173: Kindly provide the reason for the PCA analysis. Was it used to select the traits used for clustering? if so, please it should come before cluster analysis.

Line 257: you cited S5 Table before S3 Table in line 261 and I think the order should be reversed since it is their first citation in the manuscript

Line 370: I think the discussion is too long and contain many redundant statements. Endeavour to reduce the discussion and be more precise.

**Do you want your identity to be public for this peer review?** For information about this choice, including consent withdrawal, please see our Privacy Policy

Reviewer #2: **Yes: ** Massoud Dehdari

Reviewer #3: **Yes: ** Adejumobi Idris Ishola (PhD)

---

## [Author Response · Author response to Decision Letter 3]

30 Aug 2025

2-For yield, D50F and D95M traits, you should introduce the superior genotype at each location because the interaction has become significant.

Response: We appreciate the reviewer’s valuable comment. We would like to respectfully note that this point was addressed in our previously submitted revised manuscript. Specifically, in the Results section (lines 252–260 and S5 Table), we have already presented the superior genotypes for grain yield, days to 50% flowering (D50F), and days to 95% maturity (D95M) at each location, in response to this earlier suggestion. We have double-checked the revised version to ensure this information is clearly included.

Table2: what does the SNP designation in the Table represent?

Response: We thank the reviewer for raising this point. In Table 2, two sets of codes are presented for the same soybean genotypes. The Genotype Code (G01–G45) was assigned for use in the phenotypic data analysis to simplify reference to the long pedigrees. The SNP Designation (SY001–SY045) refers to the corresponding codes used for the same genotypes in the SNP-based genetic diversity analysis. Both codes, therefore represent the same set of genotypes but were applied in different parts of the study.

---

## [Decision Letter · Decision Letter 3]

8 Sep 2025

Genetic diversity and population structure analysis of soybean [Glycine max (L.) Merrill] genotypes based on agro-morphological traits and SNP markers

PONE-D-25-08250R3

Dear Dr. Abebawork Tilahun Assfaw,

We’re pleased to inform you that your manuscript has been judged scientifically suitable for publication and will be formally accepted for publication once it meets all outstanding technical requirements.

Kind regards,

Yêyinou Laura Estelle Loko

Academic Editor

PLOS ONE

Reviewers' comments:

Reviewer's Responses to Questions

**Comments to the Author**

Reviewer #2: All comments have been addressed

Reviewer #3: All comments have been addressed

2. Is the manuscript technically sound, and do the data support the conclusions?

Reviewer #2: Yes

Reviewer #3: Yes

3. Has the statistical analysis been performed appropriately and rigorously?

Reviewer #2: Yes

Reviewer #3: Yes

4. Have the authors made all data underlying the findings in their manuscript fully available?

Reviewer #2: Yes

Reviewer #3: Yes

5. Is the manuscript presented in an intelligible fashion and written in standard English?

Reviewer #2: Yes

Reviewer #3: Yes

Reviewer #2: I think your manuscript is now ready for publication in the PLOS One journal. I wish you success in your future research.

Reviewer #3: (No Response)

**Do you want your identity to be public for this peer review?** For information about this choice, including consent withdrawal, please see our Privacy Policy

Reviewer #2: **Yes: ** Masoud Dehdari

Reviewer #3: **Yes: ** Idris Ishola Adejumobi (PhD)

---

## [Editor Report · Acceptance letter]

PONE-D-25-08250R3

PLOS ONE

Dear Dr. Assfaw,

I'm pleased to inform you that your manuscript has been deemed suitable for publication in PLOS ONE. Congratulations! Your manuscript is now being handed over to our production team.

Kind regards,

on behalf of

Dr. Yêyinou Laura Estelle Loko

Academic Editor

PLOS ONE